# 5-Aza-2′-Deoxycytidine (5-Aza-dC, Decitabine) Inhibits Collagen Type I and III Expression in TGF-β1-Treated Equine Endometrial Fibroblasts

**DOI:** 10.3390/ani13071212

**Published:** 2023-03-30

**Authors:** Joana Alpoim-Moreira, Anna Szóstek-Mioduchowska, Magda Słyszewska, Maria Rosa Rebordão, Dariusz J. Skarzynski, Graça Ferreira-Dias

**Affiliations:** 1CIISA—Center for Interdisciplinary Research in Animal Health, Faculty of Veterinary Medicine, University of Lisbon, 1300-477 Lisbon, Portugal; 2Associate Laboratory for Animal and Veterinary Sciences (AL4AnimalS), 1300-477 Lisbon, Portugal; 3Institute of Animal Reproduction and Food Research, Polish Academy of Sciences, 10-748 Olsztyn, Poland; 4Polytechnic of Coimbra, Coimbra Agriculture School, 3045-601 Coimbra, Portugal; 5Department of Animal Reproduction with Large Animal Clinic, Faculty of Veterinary Medicine, University of Environmental and Live Sciences, 50-366 Wrocław, Poland

**Keywords:** endometrosis, mare, collagen, epigenetics, DNMTs, fibroblasts, 5-aza-dC, demethylating inhibitor

## Abstract

**Simple Summary:**

Endometrosis is a major cause of infertility in mares and involves the excessive deposition of extracellular matrix in the mare’s endometrium, such as collagen and α smooth muscle actin (α-SMA). Collagen is formed by activated fibroblasts, which are mainly stimulated by transforming growth factor β1 (TGF-β1). Alterations in fibroblast phenotype are linked with epigenetic alterations. Unlike genetic alterations, epigenetic alterations are changes in gene function without DNA nucleotide sequence modification. Epigenetic changes can be reversed and are therefore extremely promising for therapeutic use. DNA methylation analysis is one of the most used methods to detect epigenetic changes. It can be assessed by measuring DNA methylating enzymes (*DNMT1*, *DNMT3A,* and *DNMT3B*). Thus, the aims of this study were to investigate the in vitro epigenetic regulation of mare endometrial fibrogenesis through DNMTs transcription and the effect of the epigenetic inhibitor 5-aza-2′-deoxycytidine (5-aza-dC or decitabine) on collagen expression in mare endometrial fibroblasts challenged with TGF-β1. It was observed that TGF-β1 upregulated *DNMT3A*, *COL*s, and *α-SMA* transcripts and COLs secretion. The increase in *DNMT3A* and COLs (transcripts and protein) after TGF-β1 stimulation of equine endometrial fibroblasts was reduced after treatment with the demethylating agent 5-aza-Dc, suggesting an epigenetic regulation of mare endometrial fibrosis.

**Abstract:**

Endometrosis negatively affects endometrial function and fertility in mares, due to excessive deposition of type I (COL1) and type III (COL3) collagens. The pro-fibrotic transforming growth factor (TGF-β1) induces myofibroblast differentiation, characterized by α-smooth muscle actin (α-SMA) expression, and collagen synthesis. In humans, fibrosis has been linked to epigenetic mechanisms. To the best of our knowledge, this has not been described in mare endometrium. Therefore, this study aimed to investigate the in vitro epigenetic regulation in TGF-β1-treated mare endometrial fibroblasts and the use of 5-aza-2′-deoxycytidine (5-aza-dC), an epigenetic modifier, as a putative treatment option for endometrial fibrosis. Methods and Results: The in vitro effects of TGF-β1 and of 5-aza-dC on DNA methyltransferases (*DNMT1*, *DNMT3A,* and *DNMT3B*), *COL1A1*, *COL3A1*, and *α-SMA* transcripts were analyzed in endometrial fibroblasts, and COL1 and COL3 secretion in a co-culture medium. TGF-β1 upregulated *DNMT3A* transcripts and collagen secretion. In TGF-β1-treated endometrial fibroblasts, DNA methylation inhibitor 5-aza-dC decreased collagen transcripts and secretion, but not *α-SMA* transcripts. Conclusion: These findings suggest a possible role of epigenetic mechanisms during equine endometrial fibrogenesis. The in vitro effect of 5-aza-dC on collagen reduction in TGF-β1-treated fibroblasts highlights this epigenetic involvement. This may pave the way to different therapeutic approaches for endometrosis.

## 1. Introduction

Endometrosis is responsible for infertility in mares and is characterized by excessive deposition of collagen (COL) in the endometrium, with collagen type I (COL1) and type III (COL3) being the most abundant. The periglandular deposition of collagen in the endometrium contributes to the formation of fibrotic nests, which may impair glandular flow secretion [1,2]. Indeed, it was recently demonstrated by proteomic analysis of uterine lavage fluid that the secretion of essential proteins is affected in mares with endometrosis, due to endometrial glandular function impairment [3]. Moreover, the decreased number and area of healthy glands results in a deficient nutrient exchange between the placenta and the conceptus, and therefore may hinder its viability [4,5,6]. Therefore, all these endometrial alterations may result in pregnancy failure, delayed placental development, retarded foetal growth, or abortion [2,4,7]. Transforming growth factor β1 (TGF-β1) is over expressed in several fibrotic tissues [8,9,10,11] and induces COL production in cultured fibroblasts, regardless of their origin [10]. It not only regulates cell growth, development, and tissue remodelling, but it also participates in the pathogenesis of tissue fibrosis. In the equine endometrium, the activity of TGF-β1 is correlated with endometrosis [12,13], as it is in human endometriosis [14]. Endometriosis in women is a disease characterized by the presence of endometrial tissue within the serosa of the abdominal or pelvic cavities, which is not the same condition as equine endometrosis [15]. Nevertheless, fibrosis is consistently present in all disease forms of human endometriosis and contributes to the classic endometriosis-related symptoms of pain and infertility [16]. In tissues other than the uterus, TGF-β1 induces differentiation of many cell types into myofibroblasts. These cells are characterized by α-smooth muscle actin (α-SMA) expression and the ability to deposit excessive amounts of extracellular matrix (ECM) components. Increased expression of α-SMA in fibroblasts is therefore widely interpreted as a marker of fibroblast activation [17]. Moreover, aberrant expression of TGF-β1after injury stimulates the expression of α-SMA [18] and ECM [19] in fibroblast-like cells. Fibroblasts are key effector cells in tissue remodelling. They remain persistently activated in fibrotic diseases, resulting in the progressive deposition of ECM. Although fibroblast activation may be initiated by external factors, prolonged activation can induce an “autonomous”, self-maintaining profibrotic phenotype in myofibroblasts [20]. Accumulating evidence suggests that epigenetic alterations play a central role in establishing this persistently activated pathologic phenotype of fibroblasts [20]. Epigenetic changes, unlike genetic alterations, can be reversed and are therefore extremely promising for therapeutic use [21]. DNA methylation, considered a stable epigenetic marker, can be assessed through the action of DNA methyltransferases (DNMTs: DNMT1, DNMT3A, and DNMT3B), and commonly mediates gene repression [22,23]. The most used epigenetic treatments aim to alter either DNA methylation or histone acetylation [24,25]. To date, seven agents have been approved by the USA Food and Drug Administration for the treatment of different diseases, but many more are undergoing clinical trials [26,27,28,29,30]. DNMT inhibition is considered as an efficient approach for the prevention of DNA hypermethylation alterations [31]. The ability of DNMT inhibitors to reverse epimutations is the basis of their use as novel strategies for cancer therapy [32]. These medications act like nucleotide cytosine and incorporate themselves into DNA while it is replicating [33,34,35,36,37,38]. Demethylating agents are currently used to treat myelodysplastic syndromes (MDS) and acute myeloid leukemia (AML). They are also being experimentally used in solid tumours in low dose administration, with very promising results [39,40]. Recent studies have shown an epigenetic involvement in several human fibrotic disorders [41,42,43,44,45]. Moreover, epigenetic involvement in equine endometrial fibrosis has been demonstrated in our previous studies [46,47]. As such, we have hypothesized that endometrial fibroblasts might be under epigenetic regulation.

Therefore, our objective was to analyze the effect of TGF-β1 on methylating enzymes and the effect of the licensed demethylating agent 5-aza-2′deoxycytidine (5-aza-dC, 5-aza or decitabine) on TGF-β1-stimulated equine endometrial fibroblasts. For that purpose, we first evaluated DNA methylation through the expression pattern of *DNMT1*, *DNMT3A,* and *DNMT3B* on TGF-β1-stimulated mare fibroblasts to determine if equine endometrial fibroblasts were under epigenetic regulation. Further, we aimed to confirm whether the TGF-β1-induced alterations on COL1 and COL3 expression by equine endometrial fibroblasts could be reversed by the action of 5-aza-dC. Thus, we determined the transcription levels of *DNMT* enzymes, ECM components, and α-SMA, as well as COL1 and COL3 protein concentration, before and after fibroblasts were treated with TGF-β1, 5-aza-dC, or both.

## 2. Materials and Methods

Uteri (*n* = 5) were obtained post mortem from cyclic mares at a local abattoir (Rawicz, Poland) from April to June, according to the protocols approved by the local institutional committee for animal care and use. The mares were declared clinically healthy by independent official government veterinary inspectors. Immediately before death, peripheral blood samples were collected into heparinized tubes for subsequent progesterone (P_4_) analysis. The animals were slaughtered for meat, as part of the routine breeding and slaughter of animals, and in agreement with the European mandates (EFSA, AHAW/04-027). The internal genitalia (uteri and ovaries) were retrieved within 5 min of animal death. In this study, uteri from mares in the follicular phase of the estrous cycle were used. The follicular phase was identified based on the macroscopic observation of ovaries and P_4_ analysis of blood plasma. This phase was characterized by the absence of an active *corpus luteum* (CL) and the presence of follicles larger than 35 mm in diameter, with a concentration of P_4_ < 1 ng/mL [48]. Samples of endometria were placed in 4% buffered formaldehyde for histological examination and for endometrial categorization, according to Kenney and Doig [49].

### 2.1. Isolation and Culture of Fibroblasts

The fibroblasts isolated from healthy endometria (Kenney and Doig’s category IIA endometria) were isolated according to Szóstek-Mioduchowska et al. [50]. In the laboratory, the uterine lumen was washed three times with 10 mL of sterile Hanks’ balanced salts (HBSS; H1387; Sigma-Aldrich, Saint Louis, MO, USA) containing 0.01% of antibiotic/antimycotic (AA) solution (AA5595; Sigma Aldrich, Saint Louis, MO, USA). The uterine horns were split open with scissors to expose the endometrial surface. Endometrial strips were excised from the myometrium layer with a scalpel, washed once with sterile HBSS containing 0.01% of AA solution, and sliced with a scalpel into small fragments (1–3 mm). A single digestion of the minced tissues was performed by agitation for 45 min, in 100 mL of sterile HBSS with 0.05% (*w*/*v*) collagenase I (C2674, Sigma-Aldrich, Saint Louis, MO, USA), 0.005% (*w*/*v*) DNase I (11284932001; Roche-Sigma Aldrich, Saint Louis, MO, USA), 0.01% AA, and 0.1% (*w*/*v*) bovine serum albumin (BSA; A9418, Sigma Aldrich, Saint Louis, MO, USA). Afterwards, to extract undigested tissue fragments, filtration of the cell suspension was accomplished with 70 μm and 40 µm filters. The filtrate was mixed gently with 1 mL of Red Blood Cell Lysing Buffer Hybri-Max™ (R7757; Sigma-Aldrich, Saint Louis, MO, USA) to lyse red blood cells. Afterwards, the filtrate was washed three times by centrifugation (4 °C, 100× *g*, 10 min) in HBSS supplemented with antibiotics and 0.1% (*w*/*v*) BSA. The final pellet of endometrial cells was resuspended in FBM^TM^ Basal Medium (CC-3131, LONZA, Basel, Switzerland) supplemented with FGM^TM^-2 SingleQuots^TM^, ascorbic acid (100 ng/mL; A4544; Sigma-Aldrich, Saint Louis, MO, USA) and 0.01% of AA solution. A hemocytometer was used to perform cell count. The trypan blue exclusion test was employed to assess the viability of endometrial cells, which was higher than 95%. The immunofluorescent staining for vimentin was utilized to evaluate the homogeneity of fibroblasts, based on the protocol described [51] (Figure 1). The cells were then independently seeded, at a density of 1 × 10^5^ viable cells/mL and incubated at 38.0 °C in a 5% CO_2_ atmosphere. The medium was replaced 18 h after plating, to purify fibroblast population. At that time, the selective attachment of fibroblasts had taken place and elimination of other types of endometrial cells was possible. The medium was replaced until the cells achieved confluence, every 48 h. Fibroblast purity after isolation was approximately 96%. After reaching 90% of confluency, the cells were cryopreserved, as described previously [52].

### 2.2. Preliminary Studies

To determine the most adequate protocol to be used in our study, preliminary studies were performed. The dose of 10 ng/mL of TGF-β1 was chosen based on other studies, as the treatment has previously shown to maximally activate myofibroblasts [13,53,54]. Transforming growth factor β1 treatment of 48 h was initially chosen, based on other studies in which a maximum increase in collagen expression was achieved at that specific incubation time [13,20,54]. However, based on the analysis of preliminary studies’ results, treatment with TGF-β1 for 96 h was preferred.

To establish the appropriate dose of 5-aza-dC, a preliminary study was performed, in quadruplicate, with 0 µM, 1 µM, or 5 µM, *n* = 4. The 1 µM dose achieved the same results in collagen reduction as 5 µM (Appendix A). Therefore, the rationale was to choose the lowest concentrations due to the toxic side effects of 5-aza-dC (in clinical use) and based on other studies that achieved the same results [20,32,55]. A preliminary study was also performed to determine the protocol and the duration of 5-aza-dC treatment. The experiments were done separately, but with the same endometrial fibroblasts, in quadruplicate and with *n* = 5 for both protocols. As such, some endometrial fibroblasts were incubated with vehicle (control), 5-aza-dC (1 µM), TGF-β1 (10 ng/mL), or 1 µM 5-aza-dC + 10 ng/mL TGF-β1 at the same time for 48 h (protocol #1), and others with vehicle (control), 5-aza-dC (1 µM), TGF-β1 (10 ng/mL) for 96 h, or TGF-β1 (10 ng/mL) for 96 h + 5-aza-dC (1 µM) added after 48 h from the beginning of the experiment (protocol #2), with a total cell incubation time of 96 h. With protocol #1, no reduction was observed in COL1 and COL3 mRNA levels and protein expression or α-SMA mRNA levels in fibroblasts incubated with 1 µM 5-aza-dC and 10 ng/mL TGF-β1 simultaneously, for 48 h (Appendix A). In contrast, in protocol #2, both mRNA and protein concentration of COLs were reduced. Hence, protocol #2 was chosen and is presented here, since it also mimics the clinical conditions in which it may be used (treatment after fibrosis development and not its prevention). The duration of 5-aza-dC treatment was chosen as the minimum time (48 h) at which positive results were achieved in other studies [54,55].

The effect of TGF-β1 and of 5-aza-dC on cell viability was also analysed. After reaching confluence in a 96-well plate, fibroblasts were exposed to TGF-β1 (10 ng/mL) or 5-aza-dC (1 µM or 5 µM) for 48 h. Fibroblast viability was measured using In Vitro Toxicology Assay Kit, MTT based (TOX-1KT, Sigma Aldrich, Madison, WI, USA). None of the doses used showed toxic effects (Appendix A).

### 2.3. Treatment of Cultured Fibroblasts

Thawed fibroblasts were seeded at a density of 1 × 10^5^ viable cells/mL on T75 cm^2^ cell culture flasks. After reaching 90% confluence, fibroblasts were seeded on 24-well plates. When fibroblasts from passage 1 reached the desired 80% confluence for 48 h treatment, the culture medium was replaced with fresh Dulbecco’s Modified Eagle’s Medium/Nutrient Mixture F-12 Ham (DMEM/Ham’s F-12; D2906; Sigma-Aldrich, Saint Louis, MO, USA) supplemented with 0.01% of AA solution, 0.1% (*w*/*v*) BSA, and ascorbic acid (100 ng/mL). The cells were incubated at 38.0 °C in a 5% CO_2_ atmosphere. Then, fibroblasts were treated as follows: (i) vehicle (control); (ii) with 10 ng/mL of TGF-β1; or (iii) with 1 µM of 5-aza-dC, at 37 °C, 5% CO_2_ for 96 h. Forty eight hours after the beginning of the experiment, 1 µM 5-aza-dC was added to TGF-β1 group for another 48 h, under the same conditions. Since the half-life of 5-aza-dC is very short, the medium was changed every 24 h and fresh 5-aza-dC was added daily for 48 h. Fibroblasts were incubated alone (control), with 1 µM 5-aza-dC or 10 ng/mL TGF-β1, for 48 and 96 h, as controls. The cells and the conditioned media were collected at 48 h and 96 h and stored at −80 °C. The collection of conditioned media, for ECM determination, was performed with 1.5 mL tubes. The disruption of cells, after incubation, was performed with 1 mL of lysis buffer RTL (1015750; Qiagen GmbH, Hilden, Germany) and stored at −80 °C for RNA extraction and PCR.

### 2.4. Total RNA Isolation, cDNA Synthesis and qPCR

Total RNA was extracted using Qiagen RNeasy^®^ mini kit (74104; QIAGEN, GmbH, Hilden, Germany) according to the manufacturer’s information, including a DNase digestion step, and the samples were stored at −80 °C. The concentration of RNA was assessed spectrophotometrically, and its quality by agarose gel electrophoresis. The absorbance ratio at 260 and 280 nm (A260/280) was approximately 2. The QuantiTect Rev. Transcription Kit (no. 205313; QIAGEN, GmbH, Hilden, Germany) was used to perform the reverse transcription of RNA (1 mg) into cDNA, following the manufacturer’s instructions, and stored at −80 °C. The ABI Prism 7900 sequence detection system with 384-well plates with SYBR Green PCR master mix (Applied Bio-systems, Foster City, CA, USA) was used to perform real-time PCR. The amplified genes were α*-SMA*, *COL1A1*, *COL3A1*, *DNMT1*, *DNMT3A,* and *DNMT3B.* Specific primers and the reference gene were designed (Table 1) using the Internet-based program Primer-3 [56] and Primer Premier software (Premier Biosoft Interpairs). *SDHA* was chosen as the most stable internal control gene, among four validated reference genes, as described [57]. All primers were manufactured by Sigma-Aldrich (Saint Louis, MI, USA). The total reaction volume (10 mL) was composed of 3 mL cDNA (1 ng), 1 mL of forward and 1 mL of reverse primers (500 nM), and 5 mL of SYBR Green PCR master mix. Real-time PCR was performed by initial denaturation (2 min at 50 °C; 10 min at 95 °C), followed by 42 cycles of denaturation (15 s at 95 °C) and annealing (1 min at 60 °C). Then, after each PCR reaction, the melting curves were achieved by gradual increases in temperature from 60 °C to 95 °C to guarantee single-product amplification. Agarose gel (2%) electrophoresis was performed to confirm product specificity. To quantify relative mRNA expression levels, data were analyzed using the equation [1/(1þE)Ct] described by Zhao and Fernald [58], where the average cycle threshold (Ct) of each sample was related to the primer efficiency (E). Transcription of the target gene was normalized to that of the reference gene and relative expression values were calculated. Relative mRNA levels of control samples were compared with treated fibroblasts data.

### 2.5. Collagen Protein Quantification

ELISA techniques were performed to quantify COL1 and COL3 concentrations, in conditioned media from cultured cells. The determination of COL1 concentration in conditioned medium was accomplished by Enzyme-Immunosorbent Assay Kit for Collagen Type I (COL1) (SEA571Eq; Cloud-Clone Corp., Katy, TX, USA). The standard curve for COL1 ranged from 3.12 to 200 ng/mL. The average of intra- and inter-assay coefficients of variation (CVs) were 11.5% and 9%, respectively. The determination of COL3 in conditioned media was accomplished by Enzyme-Immunosorbent Assay Kit for Collagen Type III (COL3) (SEA176Eq; Cloud-Clone Corp., Katy, TX, USA). The standard curve for COL3 ranged from 1.56 to 100 ng/mL. The average of intra- and inter-assay CVs were 10% and 9%, respectively. The concentrations of COL1 and COL3 in control samples were compared with treated fibroblasts data, in conditioned medium.

### 2.6. Statistical Analysis

Data are shown as the mean ± SEM. For each analysis, the Gaussian distribution of results was tested using the Shapiro and Wilk normality test (GraphPad Software version 9; GraphPad, San Diego, CA, USA). Significance was considered when *p* < 0.05. A two-way ANOVA was performed to analyse the effect of time (48 and 96 h), treatment (TGFB or 5-aza-dC), and their interaction (time × treatments) on *COL1A1*, *COL3A1*, *α-SMA*, *DNMTs* genes, and COL1 and COL3 protein expression. One-way analysis of variance (ANOVA) followed by Tukey’s multiple comparison test was used to analyse the effect of different treatments on the expression of the same genes (*COL1A1*, *COL3A1*, *α-SMA*, *DNMTs*) and COL1 and COL3 proteins at 96 h. The treatment groups were analyzed with respective control. As such, the treated groups 5-aza-dC, TGF-β1, and TGF-β1 + 5-aza-dC were compared to untreated fibroblasts (control C). In addition, TGF-β1 + 5-aza-dC treatment was compared to TGF-β1, as previously used in other studies [59,60].

## 3. Results

### 3.1. TGF-β1 Upregulated Collagen Type I, III and α-SMA Expression in Endometrial Fibroblasts

The effects of treatment time, TGF-β1 treatment, and their interaction (time × treatments) were studied. Significant interactions between time and TGF-β1 treatment were observed for *COL3A1* (*p* = 0.0008) and *α-SMA* (*p* = 0.0213) mRNA levels and for COL3 protein abundance (*p* = 0.0035). No interactions were found either for *COL1A1* transcripts (*p*= 0.1078) or COL1 protein concentrations (*p* = 0.2443). Treatment of endometrial fibroblasts with TGF-β1 increased mRNA levels of *COL1A1*, *COL3A1*, and *α-SMA* at 96 h (*p* < 0.001, *p* < 0.0001, and *p* < 0.01, respectively) (Figure 2A–C). It also increased protein concentration of COL1 at 48 h and 96 h (*p* < 0.01 and *p* < 0.0001, respectively) (Figure 2D) and of COL3 only at 96 h (*p* < 0.0001) (Figure 2E). There was a rise in *COL3A1*, *α-SMA* mRNA levels, and COL3 protein concentration between 48 h and 96 h (*p* < 0.001, *p <* 0.01, and *p <* 0.01, respectively) (Figure 2B–E). The same was not observed in *COL1A1* mRNA levels or COL1 protein concentrations (Figure 2A,D).

### 3.2. TGF-β1 Upregulated DNMT3A Expression in Endometrial Fibroblasts

We examined *DNMT1*, *DNMT3A* and *DNMT3B* gene mRNA levels in TGF-β1-treated endometrial fibroblasts to determine whether *DNMTs* regulate the collagen expression through DNA methylation, at 48 h and 96 h. TGF-β1 upregulated *DNMT3A* at 48 h and 96 h (*p* < 0.05 and *p* < 0.01, respectively) (Figure 3B), while no differences were observed for *DNMT1* or *DNMT3B* mRNA levels (Figure 3A,C). There were also no differences in *DNMT*s mRNA levels between 48 h and 96 h (*p* > 0.05). No interactions were found between time and TGF-β1 treatment for any of the *DNMTs* transcripts (*DNMT1*, *p* = 0.37; *DNMT3A*, *p* = 0.19; *DNMT3B*, *p* = 0.36).

### 3.3. 5-aza-dC Downregulated Collagen Type I and III Expression Induced by TGF-β1 in Endometrial Fibroblasts

Demethylating DNMT inhibitor 5-aza-dC was used to test whether epigenetic regulation is involved in collagen expression. Endometrial fibroblasts were treated with 5-aza-dC, TGF-β1, or TGF-β1 + 5-aza-dC. TGF-β1 upregulated *COL1A1* and *COL3A1* mRNA levels (*p* < 0.01 and *p* < 0.001, respectively). The administration of 5-aza-dC to the TGF-β1-treated fibroblasts (TGF-β1 + 5-aza-dC) downregulated their expression (*p* < 0.001 and *p* < 0.01, respectively) (Figure 4A,B). The same pattern was observed for COL1 (*p* < 0.05 and *p* < 0.01, respectively) and COL3 protein concentration (*p* < 0.05 an *p* < 0.01) (Figure 4D,E). However, the TGF-β1-induced increase of α-SMA mRNA levels in endometrial fibroblasts was not reduced with the 5-aza-dC treatment (Figure 4C) (*p* > 0.05). There was also a reduction in COL1 protein after fibroblast treatment with TGF-β1 + 5-aza-dC, with respect to non-treated fibroblasts (control—C) (*p* < 0.05; Figure 4D).

### 3.4. 5-aza-dC Downregulated DNMT3A Expression in TGF-β1-Treated Endometrial Fibroblasts

After administration of 5-aza-dC to the TGF-β1-treated fibroblasts, a decrease was observed in *DNMT3A* mRNA levels (*p* < 0.001) (Figure 5B). No alterations were found either on *DNMT1* or *DNMT3B* mRNA levels after administration of 5-aza-dC to the TGF-β1-treated fibroblasts (Figure 5A,C).

### 3.5. 5-aza-dC Down-Regulated COL1A1 and Upregulated α-SMA Expression at 96 h

The effects of treatment time, 5-aza-dC treatment, and their interaction (time × treatments) were also studied. Significant interactions between time and 5-aza-dC treatment were observed for *COL1A1* (*p* = 0.0388) and *COL3A1* (*p* = 0.0078) mRNA levels. No interactions were found for *α-SMA* transcripts (*p* = 0.0746), COL1, or COL3 protein concentrations (*p* = 0.0623; *p* = 0.5382, respectively).

We also aimed to study the effect of demethylating 5-aza-dC in endometrial fibroblasts. When administered alone, 5-aza-dC downregulated *COL1A1* at 96 h (*p* < 0.05) and upregulated *α-SMA* mRNA levels at 96 h (*p* < 0.01) (Figure 6A,C), while no differences were observed for *COL3A1* mRNA (Figure 6B). Regarding COL1 and COL3 protein concentrations, only COL1 increased at 48 h (*p* < 0.05), while no other differences were found at 96 h or in COL3 (Figure 6D,E). There was a decrease in *COL1A1* and a rise in *COL3A1* transcripts between 48 h and 96 h (*p* < 0.05; *p <* 0.01 and *p <* 0.01, respectively) (Figure 6A,B).

### 3.6. 5-aza-dC Downregulated DNMT1, DNMT3A and DNMT3B Expression

The effects of treatment time, 5-aza-dC treatment, and their interaction (time × treatments) in the mRNA levels of methylating enzymes *DNMT1*, *DNMT3B*, and *DNMT3A* were also studied. Significant interactions between time and 5-aza-dC treatment were observed for *DNMT1* (*p* = 0.0006), *DNMT3A* (*p* = 0.0342), and *DNMT3B* (*p* = 0.0161) mRNA levels and for COL3 protein abundance (*p* = 0.0035).

There was a decrease in mRNA levels of *DNMT1* and *DNMT3B* at 96 h (*p* < 0.01; *p* < 0.001, respectively) and of *DNMT3A* at 48 h (*p* < 0.05) (Figure 7A–C). An increase of *DNMT3A* mRNA levels (*p* < 0.01) and a decrease of *DNMT1* and *DNMT3B* mRNA levels between 48 h and 96 h treatment (*p* < 0.001) were also observed (Figure 7A–C).

### 3.7. TGF-β1 Induced Collagen Type I and III, and DNMT3A Expression in Equine Endometrial Fibroblasts and These Effects Were Reversed by 5-aza-dC

There was an increase in mRNA levels of *DNMT3A* in TGF-β1-treated endometrial fibroblasts (*p* < 0.05; Figure 8C), as well as a rise in collagen type I and III mRNA levels (*p* < 0.01; *p* < 0.001, respectively; Figure 8A,B)), and COL1 and COL3 protein concentration (*p* < 0.001 and *p* < 0.01, respectively; Figure 8D,E). After 5-aza-dC treatment of TGF-β1-induced fibroblasts, a reduction in *DNMT3A* mRNA (*p* < 0.001) was observed (Figure 8C), simultaneously with a decrease in collagen type I and III transcripts (*p* < 0.01; Figure 8A,B) and COL1 and COL3 protein concentrations (*p* < 0.0001 and *p* < 0.001, respectively; Figure 8D,E).

A study between all control groups at different time points was performed for all the mentioned genes, to determine if the observed changes could be happening without any treatment. However, no differences were observed for any of the genes under study during the different incubation time periods (hours).

## 4. Discussion

We have previously reported increased *DNMT3B* mRNA levels in equine endometrial fibrosis [46], and increased concentrations of COL1 and COL3 proteins with the degree of endometrosis [61]. We have also demonstrated epigenetic modulation of equine endometrial fibrosis by hypermethylation of the promoter region of the anti-fibrotic *MMP2* and *MMP9* genes [47]. Thus, to evaluate what was taking place at the cellular level, we aimed to study the epigenetic mechanisms associated with TGF-β1 action in endometrial fibroblasts. Regulation of collagen expression has been extensively studied and a plethora of evidence has indicated that TGF-β1 is an important regulator of ECM metabolism in different organs [62].

Our findings indicate that in equine endometrial fibroblasts, TGF-β1 upregulated the expression of collagen type I, collagen type III, and *α-SMA* mRNA levels and COL1 and COL3 secretion at 96 h, and only COL1 secretion at 48 h. The same increase in *COL1A1* mRNA has been observed in TGF-β-treated human fibroblasts from a dermal cell line [63]. Moreover, TGF-β1 can also facilitate epithelial-mesenchymal transition (EMT) by downregulating cell junction expressions and facilitating cell motility and proliferation of human endometrial cells [14,64].

Furthermore, TGF-β1 induced an increase in *DNMT3A* expression at 48 h and 96 h. This agrees with other studies on humans, which have shown increased DNMTs expression in lung fibrosis [65], cardiac fibroblasts [66], and skin fibroblasts from systemic sclerosis patients [20]. In addition, in human nasal epithelial cells, *DNMT3A* might be the most affected by TGF-β1 [67], as happened in our study. It was suggested in the same study that DNMT inhibitors suppress the progression of chronic rhinosinusitis pathology by regulating DNA methylation. In primary mouse renal fibroblasts, DMNT1 expression was induced by TGF-β1 [68]. Nevertheless, in contrast, TGF-β1 downregulated *DNMT1* and *DNMT3A* and upregulated *COL1A1* mRNA expression and COL1 secretion in cardiac fibroblasts [54]. Interestingly, treatment with 5-aza-dC abrogated the effects of TGF-β1-induced myofibroblasts in human cardiac cells [69].

These different results may be explained by the passage number of the cells used in each study. Moreover, it may also be ascribed to the type of cells used (cardiac cells vs. renal cells), since it has been established that methylation patterns vary in different tissues and individuals [54,70]. Thus, TGF-β1 can induce both hypermethylation or hypomethylation in genes, illustrating the complexities of the pathways that control and alter methylation patterns.

To confirm if the increase of methylation, collagen, and α-SMA mRNA expression in endometrial fibroblasts after TGF-β1 treatment could be reverted by an epigenetic drug, a demethylating epigenetic modifier, 5-aza-dC, was used. The administration of 1 µg of 5-aza-dC for 48 h to TGF-β1-stimulated equine endometrial fibroblasts (for 48 h) was effective in reducing the increased COL1 and COL3 expression (mRNA levels and protein concentration) to normal levels (control), but not for *a-SMA* transcripts. The same results were observed in human cardiac fibroblasts [59]. Our data also agree with other studies on humans, where it was observed that 5-aza-dC mitigates renal [68,71], cardiac [59,72], and pulmonary [60,73,74] fibrosis by reducing hypermethylation of genes associated with fibroblast activation. In another study, it was found that hypermethylation contributes to renal fibrosis and inhibition of DNMTs suppresses chronic unilateral ureteral obstruction-induced renal fibrosis [17]. A similar reduction in the upregulation of COL1 transcripts and protein expression was reported after administration of 5-aza in TGF-β1-induced human dermal fibroblasts [20]. Furthermore, in a study on human hepatic stellate cells, it was observed that 5-aza-dC inhibited their differentiation into myofibroblasts [75]. In addition, 5-aza suppressed fibrogenic changes in human conjunctival fibroblasts [55]. On the contrary, treating TGF-β1-treated rat lung fibroblasts with 5-aza-dC stimulated *a-SMA* gene expression by inhibiting DNMTs [32].

In our study, the administration of 5-aza-dC alone to equine endometrial fibroblasts for 48 h decreased *DNMT3A* mRNA levels, but not *DNMT1* or *DNMT3B* transcripts. When fibroblasts were exposed to 5-aza-dC for 96 h, it reduced *DNMT1* and *DNMT3B* mRNA levels, but not *DNMT3A*. Neveu et al. [60] also reported a decrease in *DNMT1* expression after treatment with 5-aza-dC alone in lung fibroblasts. In our study, there was a reduction of *COL1A1* mRNA levels and an increase of *a-SMA* mRNA levels at 96 h after the treatment of endometrial fibroblasts with 5-aza-dC alone. However, COL1 protein concentration increased at 48 h, but no difference was found at 96 h, despite its decreased mRNA levels at 96 h. As a demethylating agent, 5-aza-dC may have led to hypomethylation of the *a-SMA* gene, thereby increasing its expression. The reason for *COL1A1* mRNA levels decreasing after exposure to 5-aza-dC (96 h) is not clear. There are many contradictory results regarding the effect of 5-aza-dC on ECM components. Some studies have shown an upregulation of *COL1A1* mRNA levels after treatment with 5-aza-dC alone [54], while others found no differences [59,60]. This might be explained by different experimental conditions and/or types of cells. However, further investigation is needed.

In the present study, the endometrial fibroblasts were challenged with TGF-β1 for 48 h and then treated with 5-aza-dC for a further 48 h, so the treated group was not exposed to 5-aza-dC for the total duration of the experiment (96 h). It appears that a longer exposure to 5-aza-dC might provoke bigger changes in both *DNMT*s, COLs, and *a-SMA* expression. However, the 48-h incubation period was enough to reduce the increased collagen expression induced by TGF-β1 in endometrial fibroblasts, although the same did not happen for *a-SMA.* Due to the toxic effects of demethylating agents, the minimal dose that produced the desired effect was used as a rationale for future clinical trials. Some limitations of this study include the lack of information of DNMTs and α-SMA protein expression and DNMTs activity. Likewise, soluble COL1 and COL3 were only assessed in the culture medium by ELISA, and no immunoblots were performed for their detection in the fibroblasts. Although the rationale for using COL1 and COL3 Elisa kits was to use equine-specific antibodies, the western blotting could help to assess changes within the ECM. The stimulation of endometrial fibroblasts with TGF-β1 increased COLs (mRNA levels and protein concentration), *a-SMA*, and *DNMT3A* mRNA levels, and the treatment with 5-aza-dC decreased their expression, except for *a-SMA* gene expression, suggesting an epigenetic regulation through the alteration of DNA methylation.

In summary, although epigenetic alterations have been implicated in the development of many types of cancer [76,77], the role of epigenetic changes in fibrosis, particularly in equine endometrial fibrosis, needs much further investigation, since this research is relatively preliminary. Based on our findings, along with the evidence of other studies linking DNA methylation and fibrosis, one may suggest that DNA methylation plays a role in the pathogenesis of endometrial fibrosis. Therefore, pharmacological modulation of this process may result in an effective treatment for endometriosis.

## 5. Conclusions

The increase in *DNMT3A* and COLs (mRNA and protein) after TGF-β1 stimulation of equine endometrial fibroblasts was reduced after treatment with a demethylating agent (5-aza-dC), suggesting an epigenetic regulation of mare endometrial fibrosis.

## Figures and Tables

**Figure 1 animals-13-01212-f001:**
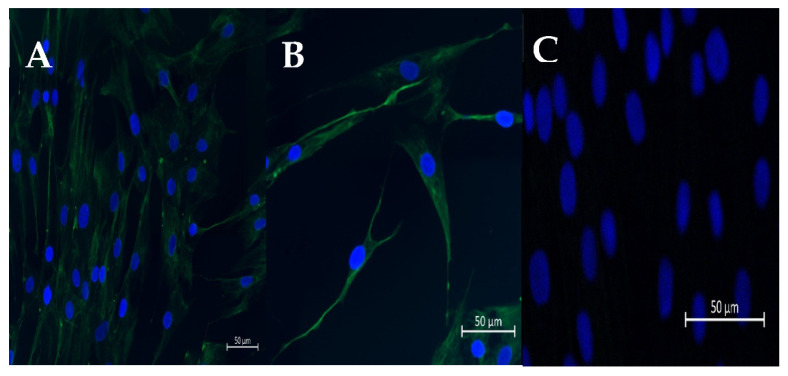
Representative pictures of immunofluorescence staining of vimentin (**A**,**B**) in cultured fibroblasts. (**C**) DAPI staining. Scale bar = 50 μm (magnification: 20,40).

**Figure 2 animals-13-01212-f002:**
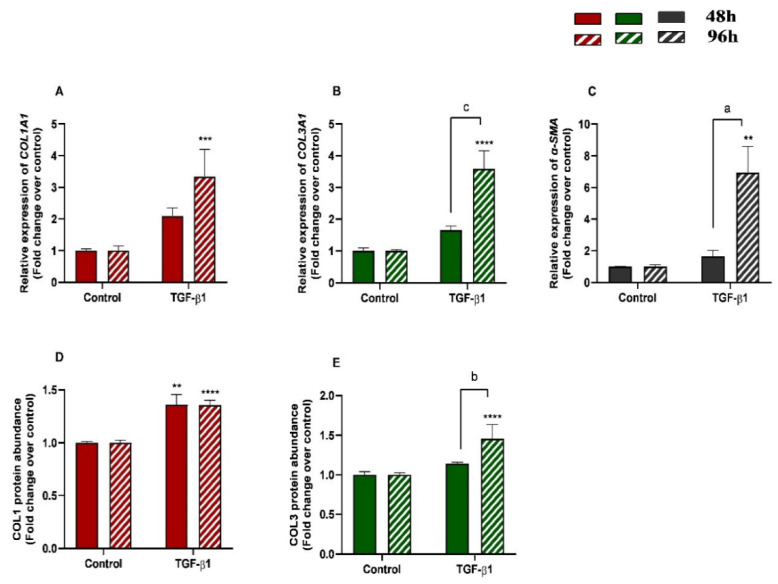
Relative levels of (**A**) collagen type I (*COL1A1*), (**B**) collagen type III (*COL3A1*), and (**C**) α-smooth muscle actin (*α-SMA*) mRNA, and of (**D**) COL1 and (**E**) COL3 protein concentrations in mare endometrial fibroblasts after incubation with TGF-β1 (10 ng/mL) for 48 h and 96 h; *n* = 5. All values are expressed as percentage of change from respective hour control (non-treated fibroblasts). Bars represent mean ± SEM. Asterisks indicate significant differences between treatment and the respective control (** *p <* 0.01, *** *p <* 0.001, **** *p <* 0.0001); a and b letters indicate significant differences between treatment hours (a—*p* < 0.05; b—*p* < 0.01; c—*p* < 0.001).

**Figure 3 animals-13-01212-f003:**
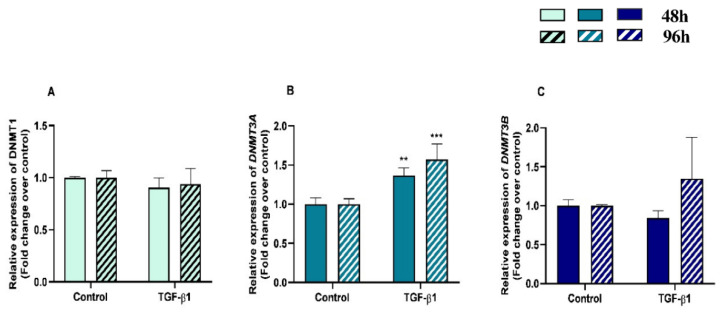
Relative DNA methyltransferases ((**A**)—*DNMT1*, (**B**)—*DNMT3A*, and (**C**)—*DNMT3B*) mRNA levels in mare endometrial fibroblasts after incubation with TGF-β1 (10 ng/mL) for 48 h and 96 h; *n* = 5. All values are expressed as percentage of change from respective hour control (non-treated fibroblasts). Bars represent mean ± SEM. Asterisks indicate significant differences from the respective control (** *p <* 0.01, *** *p <* 0.001).

**Figure 4 animals-13-01212-f004:**
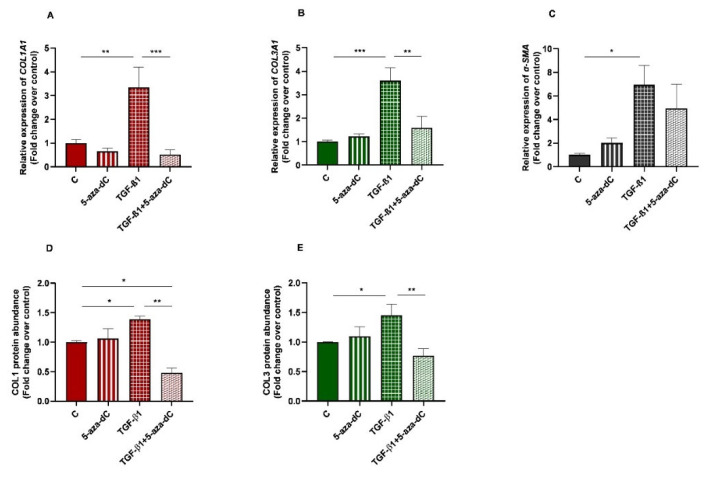
Relative levels of (**A**) collagen type I (*COL1A1*), (**B**) collagen type III (*COL3A1*), and (**C**) α-smooth muscle actin (*α-SMA*) mRNA, and of (**D**) COL1 and (**E**) COL3 protein concentrations in non-treated (control-C) endometrial fibroblasts or treated with 5-aza-dC (1 µM), TGF-β1 (10 ng/mL) for 96 h or both combined (TGF-β1 followed by 5-aza-dC at 48 h—total cell incubation of 96 h); *n* = 5. Each treatment was compared to respective control (all groups with control C and TGF-β1 + 5-aza-dC with TGF-β1). Bars represent mean ± SEM. Asterisks indicate significant differences between treatments (* *p <* 0.05, ** *p <* 0.01, *** *p <* 0.001).

**Figure 5 animals-13-01212-f005:**
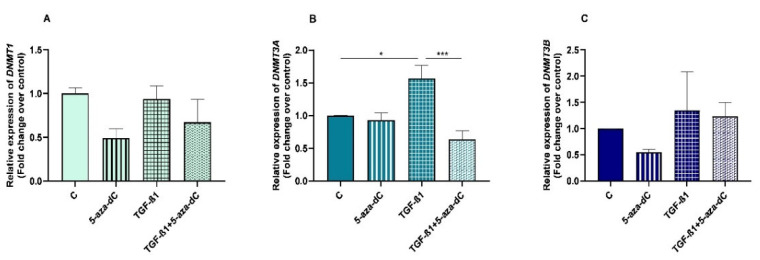
Relative DNA methyltransferases ((**A**)—*DNMT1*, (**B**)—*DNMT3A* and (**C**)—*DNMT3B*) mRNA levels in non-treated (control-C) mare endometrial fibroblasts or treated with 5-aza-dC (1 µM), TGF-β1 (10 ng/mL) for 96 h or both combined (TGF-β1 followed by 5-aza-dC at 48 h—total cell incubation of 96 h); *n* = 5. Each treatment was compared to respective control (all groups with control C and TGF-β1 + 5-aza-dC with TGF-β1); *n* = 5. Bars represent mean ± SEM. Asterisks indicate significant differences between treatments (* *p <* 0.05, *** *p <* 0.001).

**Figure 6 animals-13-01212-f006:**
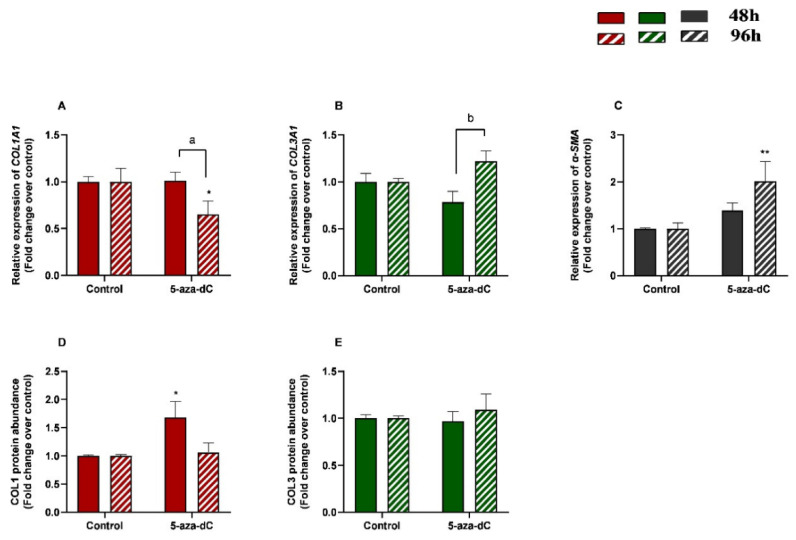
Relative levels of (**A**) collagen type I (*COL1A1)*, (**B**) collagen type III (*COL3A1*), and (**C**) α-smooth muscle actin (*α-SMA*) mRNA, and of (**D**) COL1 and (**E**) COL3 protein concentrations in mare endometrial fibroblasts after incubation with TGF-β1 (10 ng/mL) for 48 h and 96 h; *n* = 5. All values are expressed as percentage of change from respective hour control (non-treated fibroblasts). Bars represent mean ± SEM. Asterisks indicate significant differences between treatment and the respective control (* *p <* 0.05; ** *p <* 0.01); a and b letters indicate significant differences between treatment hours (a—*p* < 0.05; b—*p* < 0.01).

**Figure 7 animals-13-01212-f007:**
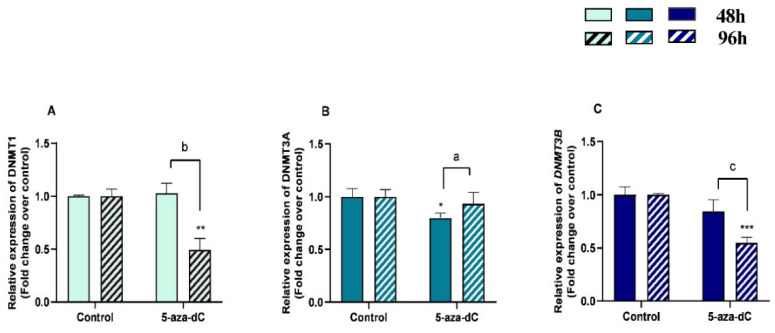
Relative DNA methyltransferases ((**A**)—*DNMT1*, (**B**)—*DNMT3A*, and (**C**)—*DNMT3B*) mRNA levels in mare endometrial fibroblasts after incubation with 5-aza-dC (1 µM) for 48 h and 96 h; *n* = 5. All values are expressed as percentage of change from respective hour control (non-treated fibroblasts). Bars represent mean ± SEM. Asterisks indicate significant differences from the respective control (* *p* < 0.05, ** *p <* 0.01, *** *p <* 0.001). a, b, and c letters indicate significant differences between treatment hours (a—*p* < 0.05; b—*p* < 0.01; c—*p* < 0.001).

**Figure 8 animals-13-01212-f008:**
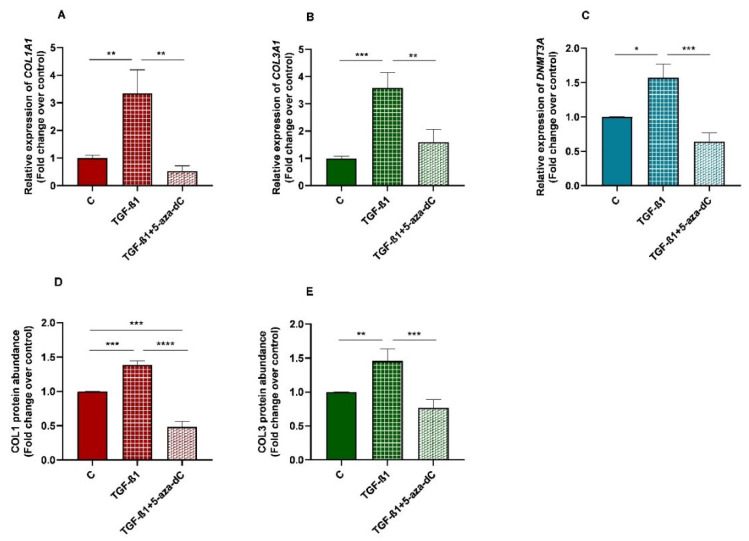
Relative levels of (**A**) collagen type I (*COL1A1*), (**B**) collagen type III (*COL3A1*), and (**C**) DNA methyltransferase 3A (*DNMT3A*) mRNA, and of (**D**) COL1 and (**E**) COL3 protein concentrations in non-treated (control-C) endometrial fibroblasts or treated with TGF-β1 (10 ng/mL) for 96 h or TGF-β1 (10 ng/mL) + 5-aza-dC (1 µM) (TGF-β1 followed by 5-aza-dC at 48 h—total cell incubation of 96 h); *n* = 5. Bars represent mean ± SEM. Asterisks indicate significant differences between treatments (* *p <* 0.05, ** *p <* 0.01, *** *p <* 0.001, **** *p <* 0.0001).

**Table 1 animals-13-01212-t001:** Primer sequences used in qPCR study.

Gene (Acession Number)	Sequence 5′-3′		Amplicon (Base Pairs)
	Forward	Reverse	
*a-SMA* XM_001503035.6	TCAGCTTCCCTGAACACCAC	GCAAAGCCAGCCTTACAAAG	151
*COL1A1* XM_014736922.1	TAAGGGTGACAGAGGCGATG	GGACCGCTAGGACCAGTTTC	144
*COL3A1* ENSECAT00000026771.3	CAAAGGAGAGCCAGGAGCAC	CTCCAGGCGAACCATCTTTG	98
*DNMT1* (XM_023645449.1)	CAAGGCAAACAACCAGGCA	CTTCCTCCTCTTCCGTGTGTGT	237
*DNMT3A* (XM_023619394.1)	GCCTCAATGTCACCCTGGAA	AAGAGGTCCACACATTCCACG	206
*DNMT3B* (XM_023626333.1)	GAGCTGGCAAGACTTTCCCC	TTGGGTGGAGGGCAGTAGTC	198
*SDHA2* DQ402987.1	GAGGAATGGTCTGGAATACTG	GCCTCTGCTCCATAAATCG	91

Note: *a-SMA*-alpha smooth actin; *COL1A1*-collagen type 1, alpha2; *COL3A1*-collagen type III, alpha-1; *DNMT1*, *DNMT3A*, *DNMT3B*—DNA methyltransferase 1, 3A, 3B; *SDHA2*—succinate dehydrogenase flavoprotein subunit.

## Data Availability

Data will be available upon request to the corresponding author.

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
