# Peer review of "5-Aza-2′-Deoxycytidine (5-Aza-dC, Decitabine) Inhibits Collagen Type I and III Expression in TGF-β1-Treated Equine Endometrial Fibroblasts"

_animals, 2023, doi:10.3390/ani13071212_

Round 1

Reviewer 1 Report

Endometritis in mares is a common ailment in the industry and deserving of attention and therapeutic options.  The authors hypothesize that the excess collagen produced is due to TGFb possibly acting in an epigenetic manner.  Various in vitro experiments were performed on endometrial fibroblasts with and without TGFb and with and without AzaC, a demethylating agent.  Measures of collagen production and DNMT gene expression were taken over time.  From the data, the authors conclude that TGFb increases COL and DNMT3a expression that is suppressed by 5-aza C.  As written however, the paper is fatally flawed due to inappropriate design and analysis.  Specifically, 

The design is incorrect leading to inappropriate stats.  When treating +/- TGF, the control wells (-TGF) need to be evaluated at 48 and 96 hr (Figure 2).   The design becomes a 2 X 2 factorial with 2-way ANOVA statistical analysis.  In its present form, you are unable to determine if the increased gene expression is due to exogenous TGF addition or to autocrine release of a factor into the media.   This is further complicated by the fact that you did sequential incubation of cells with the growth factor and chemical which requires additional time (48/96 h) and treatment controls (Fig 4,5,8). 

The calculation Ct and analysis of fold-change in Figures 2-8 demonstrate no variation in the 0 h (no SEM) which is incorrect.  Normalization of gene expression should be delta Ct= control gene expression for the rep or isolate – average of control gene expression for the 5 replicates. 

The topic is relevant but the experimental design prevents interpretation and advancement of our understanding of endometritis.

Author Response

Please find letter in attachtment.

Reviewer 2 Report

the manuscript is interesting, generally well written and illustrated. However, the manuscript presents some points that can be improved. In particular; 

Line 59: It deserves to be pointed out that TGFB1 is also involved in human endometriosis (see PMID: 26708185).

Line 345-348: It is interesting to poin out that these results are also in agreement with the effect of TGFB1 found in human fibroblast ( see PMID: 32006713)                                                       

4. Discussion: it deserves to be mentioned that TGFB1 can facilitate epithelial-mesenchymal transition (EMT) by downregulating cell junctions expression (see PMID: 24768095) and facilitating cell motility and proliferation (see PMID: 26708185) of endometrial cells. This is an interesting point to add since it can further highlight the interesting results obtained by the authors 

An accurate revision of typing errors is recommended

Author Response

Please find letter in attachment.

Reviewer 3 Report

An elegant and interesting study investigating mechanisms of fibrosis in equine endometrial cells. Please consider the following points to improve the readability of the paper:

Introduction

Lines 53 and 54: Correct to read “… is characterized by excessive deposition of collagen (COL) in the endometrium, with collagen type I (COL1) and type III (COL3) being the most abundant.”

Lines 78-80: Consider adding a few lines about demethylating agents; what are their mechanisms, add citations about their utility, etc.

Consider providing more background about the roles of COL1, COL3, alpha-SMA in fibrosis and fibroblast biology.

Methods

Line 148-149: It would be nice to have the preliminary dosing study presented with a graph if possible. How was this quantified?

Lines 153-157: If possible, it would be interesting to see graphical depictions of the differences between protocol #1 and protocol #2. Ideally, a photograph. Was this quantitatively or qualitatively assessed? These preliminary studies are interesting and can be presented a bit more thoroughly.

Line 156: Please specify here in the text what “no reduction” means… number of fibroblasts? mRNA? Protein?

It would be comprehensive if there was a no treatment control with fibroblasts that were not treated with TGF-beta1 nor 5-aza-dC and assessed at 0, 48, and 96 hours.

Results

Line 245: Correct to read “…no differences in COL1 and COL3 protein concentrations between 48 h and 96 h…”

Figures 2 and 3: To improve the clarity of the figures, I suggest moving “TGF-beta1” to the title of the figures (top). As currently displayed, it first appears that the TGF-beta1 is the dependent variable, which is inaccurate.

Figures 4 and 5: Please add information about the timing of treatment in the figure legends so that the figures can stand alone. Also, please be sure that the “n” is included in each figure legend.

Discussion

Line 341: Correct to read “…the epigenetic mechanisms associated with TGF-B1 action…”

Author Response

Please find letter in attachment

Round 2

Reviewer 1 Report

The requested edits were made. Thanks